# In Vitro Antioxidant and Anti-*Propionibacterium acnes* Activities of Cold Water, Hot Water, and Methanol Extracts, and Their Respective Ethyl Acetate Fractions, from *Sanguisorba officinalis* L*.* Roots

**DOI:** 10.3390/molecules23113001

**Published:** 2018-11-16

**Authors:** Seongdae Kim, Sung Oh, Han Byul Noh, Seongmi Ji, Song Hee Lee, Jung Mo Koo, Chang Won Choi, Hyun Pyo Jhun

**Affiliations:** 1Department of Biology & Medicinal Science, Pai Chai University, Daejeon 35345, Korea; khboy111@pcu.ac.kr (S.K.); 5star@pcu.ac.kr (S.O.); creator1018@pcu.ac.kr (H.B.N.); hijsm91@gmail.com (S.J.); thdgml6245@naver.com (S.H.L.); jungmo9@gmail.com (J.M.K.); 2Daeduck Lab. Co. Ltd., Daejeon 34024, Korea; hpjhon@daeducklab.com

**Keywords:** *Sanguisorba officinalis* L. root, cold water extract, hot water extract, methanol extract, ethyl acetate fraction, anti-*Propionibacterium acnes* activity, in vitro antioxidant activities

## Abstract

Identification of medicinal plants and naturally derived compounds as new natural antioxidant and antibacterial sources for topical acne treatment has long been important. To determine anti-*Propionibacterium acnes* activity and in vitro antioxidant activities, *Sanguisorba officinalis* L. root (SOR) was extracted with cold water (CWE), hot water (HWE), and methanol (ME), and each extract was fractionated successively with hexane, ethyl acetate (EA), and butanol to determine whether the activities could be attributed to the total phenolic, flavonoid, terpenoid, and condensed tannin contents. Pearson’s correlation coefficients were analyzed between the respective variables. The SOR CWE, HWE, ME, and their respective EA fractions showed anti-*P. acnes* activity based on the paper disc diffusion method on agar plates, minimum inhibitory concentration (MIC), and minimal bactericidal concentration (MBC). The MIC against *P. acnes* had a moderate (+) correlation with the total phenolic content, but not with the other measures. The 2,2-diphenyl-1-picrylhydrazyl (DPPH) scavenging capacity (SC) had a strong (–) correlation with the total phenolic content and a moderate (–) correlation with the total flavonoid content. The total antioxidant capacity had a strong (+) correlation with the condensed tannin content. Linoleic acid peroxidation inhibition had a strong (–) correlation with the total phenolic content. To elucidate the major active phytochemicals in the CWE-EA, HWE-EA, and ME-EA fractions, high performance liquid chromatography-ultraviolet (HPLC-UV) and ultra high performance liquid chromatography coupled with hybrid triple quadrupole time-of-flight mass spectrometry (UHPLC-QTOF-MS) were performed. The HPLC-UV analysis showed the presence of nine compounds in common (arjunic acid and/or euscaphic acid, gallic acid, kaempferol, caffeic acid, ferulic acid, tannic acid, and coumarin, quercetin). The UHPLC-QTOF-MS analysis showed the presence of nine compounds in common (gallic acid; caffeic acid; umbelliferone; arjunic acid, euscaphic acid, and/or tormentic acid; pomolic acid; rosamultic acid; and benzoic acid). When standards of the identified phytochemicals were tested against the same bacterium, quercetin, coumarin, and euscaphic acid showed antibacterial activity against *P. acnes*.

## 1. Introduction

Skin disorders, such as acne vulgaris and atopic dermatitis, are associated with inflammation and the release of free radicals, which lead to oxidative and cellular damage, and bacterial infections. Acne vulgaris affects the face, back, shoulders, and chest, which contain the largest oil glands, and contributes to an increase in sebum excretion, comedogenesis, *Propionibacterium acnes* proliferation, and inflammation [1]. *P. acnes* acts as an immunostimulator through the production of proinflammatory cytokines, which are involved in development of the inflammatory process [2]. Inhibition of *P. acnes* decreases comedone rupture into the surrounding skin and prevents acne progression. 

Topical application of therapeutic agents has been found to be more feasible than hormonal treatment and laser therapy. Available synthetic treatments using topical benzoyl peroxide and retinoid are effective for mild acne [3]. However, benzoyl peroxide can induce side effects, such as concentration-dependent irritation and uncommon contact allergy [4]. Retinoid is also limited by side effects (dryness, peeling, erythema, and irritation) and risk of photosensitivity [3]. For mild or moderate acne, topical antibiotics, such as clindamycin and erythromycin, are used as medications, however use of topical antibiotics in combination with benzoyl peroxide is recommended due to increasing antibiotic resistance [4]. Use of oral antibiotics for moderate or severe acne can have several side effects, including photosensitivity, gastrointestinal distress, and *Candida* infections [5]. Moreover, their chronic use can lead to the development of drug-resistant bacteria [6,7]. In this respect, the ingredients in topical acne treatments, particularly herbs and naturally derived compounds, have received considerable interest, because they show fewer adverse effects than synthetic agents [8].

*Sanguisorba officinalis* L. is a member of the *Rosaceae* family and is a widely distributed perennial weed worldwide [9]. Edible dried roots of *S. officinalis* L. (known as JiYu in Korea and Japan or DiYu in China) have been used as a herbal medicine for centuries. Traditionally, *S. officinalis* L. root (SOR) has been used in Far East countries for multiple purposes, including hemostasis in hypermenorrhea and internal or external bleeding, and treatment for scalds and burns, frostbite, diarrhea, chronic intestinal infections, and duodenal ulcers [10,11]. Experimentally, *Sanguisorba* species have shown anti-Alzheimer’s [9], anticancer [12,13,14,15,16], anti-inflammatory [9,17,18], antioxidant [19,20,21], antiviral [22,23], and anti-wrinkle [24] activities. These pharmacological studies have led to the isolation of several compounds, including triterpenoids, phenolic acids, flavonoids, phenylpropanoids, and polysaccharides [9,11,12,24,25,26,27,28,29].

In particular, SOR is known to be effective against skin diseases, including urticaria, eczema, and allergic dermatitis [30], and against numerous bacteria [31,32,33]. However, antibacterial activity against *P*. *acnes* has not been reported. The aim of this study was to evaluate antibacterial activity against *P. acnes*, and the in vitro antioxidant activities of SOR. To determine whether these activities can be attributed to the total phenolic, flavonoid, terpenoid, and condensed tannin contents, SOR was extracted using cold water (CWE), hot water (HWE), and methanol (ME), and the respective extracts were further fractionated successively with hexane, ethyl acetate (EA), and butanol using separating funnels. To determine the major active phytochemicals in the respective EA fractions, high performance liquid chromatography-ultraviolet (HPLC-UV) and ultra high performance liquid chromatography coupled with triple quadrupole time-of-flight mass spectrometry (UHPLC-QTOF-MS) was performed.

## 2. Results

### 2.1. Total Phenolic, Flavonoid, Terpenoid, and Condensed Tannin Contents in the CWE, HWE, ME, and their Respective EA Fractions

As shown in Table 1, selected phytochemicals varied among the extracts and their derived EA fractions. Among three different extracts, the HWE showed the highest total phenolic, flavonoid, and condensed tannin contents (60.0 mg gallic acid equivalent (GAE)/g, 30.2 mg quercetin equivalent (QE)/g, and 1.80 mg catechin equivalent (CE)/g dry powder, respectively), while the CWE showed the highest total terpenoid content of 29.10 mg/g powder. Among three different EA fractions, CWE-EA showed the highest total phenolic, flavonoid, terpenoid, and condensed tannin contents (58.1 mg GAE/g, 20.9 mg QE/g, 29.69 mg/g, and 2.15 mg CE/g dry powder, respectively).

### 2.2. Antibacterial Activities of the CWE, HWE, ME, and their Respective EA Fractions

The CWE and HWE were active only against *P. acnes* and were not active against other bacteria. However, the ME showed antibacterial activity against all tested bacteria in the paper disc diffusion assay (Appendix A). Therefore, we focused on the antibacterial activity against *P. acnes*, which was evaluated by the paper disc diffusion assay, minimum inhibitory concentration (MIC), and minimum bactericidal concentration (MBC). In the paper disc diffusion assay (Figure 1), the inhibition zone increased in a concentration-dependent manner, reaching its maximum size at a 5 mg/disc concentration for all extracts (24.0–24.5 mm) and their EA fractions (28.0–28.5 mm). The kanamycin control showed a 30.5 mm inhibition zone size against *P. acnes* (data not shown). The MIC values of the CWE, HWE, and ME were 1250, 625, and 312 μg/mL, respectively, and their MBC values were 2500, 1250, and 312 μg/mL, respectively. The MIC values of the CWE-EA, HWE-EA, and ME-EA were 312, 156, and 78 μg/mL, respectively, and their MBC values were 312, 312, and 156 μg/mL, respectively (Table 2).

### 2.3. In Vitro Antioxidant Activity of the CWE, HWE, ME, and their Respective EA Fractions

In this study, the SOR CWE, HWE, ME, and their respective EA fractions exhibited a free radical scavenging capacity (SC) in a dose-dependent manner when the 2,2-diphenyl-1-picrylhydrazyl (DPPH) radical was used as a substrate (Figure 2). The HWE exhibited the best SC_50_ value of 7.58 μg/mL, followed by the CWE (12.14 μg/mL), ME (16.74 μg/mL), CWE-EA (19.14 μg/mL), HWE-EA (35.81 μg/mL), and ME-EA (52.46 μg/mL). 

The percentages of the total antioxidant capacity of the CWE, HWE, ME, and their respective EA fractions were in a range of 80.7 to 91.7% (Figure 3a). Although the HWE-EA value was lower than that of quercetin (99.8%), no significant difference was found between the two values. The SC percentages for hydrogen peroxide for the CWE, HWE, ME, and their respective EA fractions were in a range of 51.9 to 99.6% (Figure 3b). Among the extracts and their respective EA fractions, the SC value of the ME was close to that of quercetin (99.8%). The percentage inhibition values of linoleic acid peroxidation for the CWE, HWE, ME, and their respective EA fractions were in a range of 58.3 to 79.8% (Figure 3c). The linoleic acid peroxidation inhibition value of the ME-EA was close to that of quercetin (83.4%), and no significant difference was found between the two values.

### 2.4. HPLC-UV and UHPLC-QTOF-MS Analyses

The retention times of the peaks detected from the HPLC-UV chromatograms of the CWE-EA, HWE-EA, and ME-EA were compared to those of the reference standards tentatively proposed: (**1**) euscaphic acid or arjunic acid (or both), (**3**) gallic acid, (**4**) kaempferol, (**5**) caffeic acid, (**6**) ferulic acid, (**7**) tannic acid, (**8**) coumarin, (**9**) quercetin, and (**10**) chlorogenic acid at retention times of 2.0, 4.7, 20.5, 21.0, 27.8, 29.3, 31.5, 32.7, and 34.0 min, respectively (Figure 4a). Eight peaks were identified in CWE-EA and HWE-EA, and nine peaks were identified in the ME-EA. Regardless of the EA fractions, tannic acid (a commercial form of tannin) showed the highest peak, followed by euscaphic and arjunic acids (Figure 4b–d). 

To validate the proposed nine phytochemicals, CWE-EA, HWE-EA, and ME-EA were further analyzed by UHPLC-QTOF-MS. The MS data and the tentative identification results are shown in Table 3, Figure 5, and Appendix A. The UHPLC-QTOF-MS analysis provides score, formula, intensity, accurate mass, and retention time. In the CWE-EA, twelve compounds were tentatively identified with a mass error between −0.9 ppm and −0.1 ppm. Their overall identification scores were higher than 92%, except for rosamultic acid (54%). In the HWE-EA, nineteen compounds were tentatively identified with a mass error between −2.7 ppm and 1.0 ppm, and they showed a broad range of identification scores between 94% (pomolic acid) and 51% (catechin). In the ME-EA, nineteen compounds were tentatively identified with a mass error between −1.0 ppm and 1.0 ppm, and they showed a broad range of identification scores between 97% (pomolic acid, benzoic acid) and 50% (catechin). 

Some compounds were found only in the CWE-EA (myricetin, oleic acid), HWE-EA (hyperoside, quercetin), or ME-EA (methyl gallate, *p*-coumaric acid, coumarin). However, all the EA fractions have nine compounds in common, such as gallic acid (molecular weight of 171.03), caffeic acid (molecular weight of 181.04), umbelliferone (molecular weight of 163.04), euscaphic acid (arjunic acid, tormentic acid) (molecular weight of 489.36), pomolic acid (molecular weight of 473.36), rosamultic acid (molecular weight of 487.34), and benzoic acid (molecular weight of 123.04). Three terpenoids (euscaphic acid, arjunic acid, tormentic acid) showed the same information of score, intensity, molecular mass, and retention time. 

To confirm that the identified major compounds had anti-*P. acnes* activity, nine selected standards were tested using the paper disc diffusion method. In particular, coumarin showed the strongest inhibition of *P. acnes* growth, followed by quercetin and euscaphic acid. On the other hand, quercetin, kaempferol, ferulic acid, caffeic acid, gallic acid, tannic acid, and benzoic acid showed negative results (Figure 6). 

## 3. Discussion

In previous studies, either methanol or ethanol extracts of some plants showed higher total phenolic and flavonoid contents than the HWE [34,35]. In contrast, the SOR HWE showed higher total phenolic, flavonoid, and condensed tannin contents than the CWE and ME. Hot water extraction is therefore effective in extracting bioactive phytochemical constituents. The recovery of polyphenols from plant tissues is probably influenced by their solubility during extraction, the solvent type, the degree of phenol polymerization, the phenol-other constituent interaction, and insoluble complex formation [36]. 

Contrary to the previous study on water extracts from the aerial parts or leaves of *S. officinalis* L. [33], the SOR CWE and HWE showed antimicrobial activity against *P. acnes*. It is unclear whether the anti-*P. acnes* compounds are rich in the roots but not in the aerial parts of *S. officinalis* L. Among the extracts and their respective EA fractions, the SOR ME and ME-EA showed the lowest MIC and MBC values against *P. acnes*. Furthermore, the SOR ME and ME-EA showed antimicrobial activity against numerous Gram-negative and -positive bacteria, which corresponds to earlier works [31,37,38]. It suggested that the SOR ME displayed a broad spectrum of antibacterial activity, whereas both the CWE and HWE displayed a narrow spectrum of antibacterial activity. In a previous study, high total phenolic and flavonoid contents from ethanol extract of a plant showed strong correlation with high antibacterial activity [35]. In this study, Pearson’s correlation analysis indicated that the MIC against *P. acnes* had a moderate (+) correlation with the total phenolic content (r = 0.659, *p* = 0.15), but weak correlations with the total flavonoid (r = 0.293), terpenoid (r = 0.235), and condensed tannin (r = −0.130) contents. Although we could not predict clearly whether the total flavonoid, terpenoid, and condensed tannin contents were attributed to direct antibacterial activity against *P. acnes*, specific compounds in the extracts might be responsible for the effect.

In previous studies, the DPPH radical SC and antioxidant activities showed strong correlations with the total phenolic, flavonoid, or condensed tannin contents of many plant species [39,40,41,42,43]. In this study, Pearson’s correlation analysis indicated that the DPPH SC had a strong (–) correlation with the total phenolic content (r = −0.946, *p* < 0.01) and a moderate (–) correlation with the total flavonoid content (r = −0.557) of SOR. However, the DPPH radical SC showed weak correlations with the condensed tannin (r = 0.181) and total terpenoid (r = 0.238) contents. The total antioxidant capacity had a strong (+) correlation with the condensed tannin content (r = 0.822, *p* < 0.05), but showed weak correlations with the total phenolic (r = −0.050), flavonoid (r = −0.009), and terpenoid (r = 0.211) contents. The hydrogen peroxide SC showed weak correlations with the total phenolic (r = −0.181), flavonoid (r = −0.072), terpenoid (r = −0.194), and condensed tannin (r = −0.299) contents. Linoleic acid peroxidation inhibition had a strong (–) correlation with the total phenolic content (r = −0.826, *p* < 0.05), but weak correlations with the total flavonoid (r = −0.216), terpenoid (r = −0.437), and condensed tannin (r = −0.232) contents. Taken together, the Pearson’s correlation coefficients suggest strong relationships between the DPPH SC and the total phenolic and flavonoid contents, between the total antioxidant activity and the condensed tannin content, and between linoleic acid peroxidation inhibition and the total phenolic content of SOR.

The combination of HPLC-UV and UHPLC-QTOF-MS proved an efficient method for the detection of compounds that are responsible for antimicrobial and antioxidant activities. The qualitative accuracy of the results was much increased by mutual confirmation, and false positive results were excluded. The HPLC-UV detected tannic acid (C_76_H_52_O_46_, 1700 Da) showing the highest peak, whereas the UHPLC-QTOF-MS failed to show this compound, because the analysis covers a mass range of 100–1000 *m*/*z*. Instead, the UHPLC-QTOF-MS showed the presence of gallic acid or ellagic acid (or both) in the all EA fractions, and these hydrolysable tannins are known to form gallototannins or ellagitannins. In addition, catechin, catechin gallate, and epicatechin gallate were identified in the HWE-EA and ME-EA, and their oligomers or polymers are known to form condensed tannins (procyanidins). Recently, several tannins were used as feed additives to control diseases in poultry farms [44] due to their antibacterial activity [45,46]. 

Although ferulic acid, gallic acids [47], caffeic acid [48], and condensed tannins [49] are known to have antibacterial activity against some Gram-positive bacteria, they may not be responsible for the anti-*P. acnes* activity of SOR. The presence of such phenolics could be related to the considerable antioxidant activities (DPPH radical SC, total antioxidant capacity, hydrogen peroxide SC, and linoleic acid peroxidation inhibition) of SOR. Flavonoids are well-known antioxidants with antimicrobial properties [50], and these properties are related to their chemical structures, especially the numbers and positions of methoxyl and hydroxyl groups [29]. Chromatographic profiles also showed that quercetin and kaempferol are the most common flavonoids in all the EA fractions. In particular, quercetin in ethanol extract of tartary buckwheat bran showed strong anti-*P. acnes* activity [51]. The combination of quercetin with kaempferol showed an additive effect on antibacterial activity against *P. acnes* [52]. However, our results showed that kaempferol is not an anti-*P. acnes* compound. 

Triterpenoids, mainly 19α-hydroxyl ursolic acid (pomolic acid) derivatives and 19α-hydroxyl oleanolic acid derivatives, have been reported from SOR [25,26,53]. In the UHPLC-QTOF-MS analysis, five triterpenoids (arjunic, euscaphic, tormentic, pomolic, and rosamultic acids) were identified in all EA fractions, although arjunic, euscaphic, and tormentic acids were not separated at each peak under given HPLC conditions. Previously, euscaphic, pomolic, and tormentic acids showed antibacterial activity against *P. acnes* [54]. Besides, numerous coumarin derivatives from plants showed antibacterial activities against Gram-positive and Gram-negative bacteria [55,56]. The area of each peak is in proportion to the amount of the particular component present in the sample mixture injected into the HPLC-UV chromatography column. Based on the peak area, quercetin was more distinctive in HWE-EA than in CWE-EA and ME-EA, and coumarin was more distinctive in ME-EA than in CWE-EA and HWE-EA. Previously, methanol was known to be the best solvent to extract coumarins among the organic solvents [57], and some of them were also soluble in hot water [58]. The UHPLC-QTOF-MS supports the presence of quercetin in HWE-EA and coumarin in ME-EA. In this respect, triterpenoids combined with quercetin (HWE-EA) or coumarin (ME-EA) could be responsible for the strong anti-*P. acnes* activity. This finding explains why the MIC and MBC values of ME-EA were lower than those of CWE-EA and HWE-EA.

## 4. Materials and Methods

### 4.1. Plant Material

The plant specimen (SOR) was purchased from a local market (Korea Medicine Street, Daejeon, Korea) and deposited at the Department of Biology and Medicinal Science, Pai Chai University, Daejeon, Korea. 

### 4.2. Preparation of Cold Water, Hot Water, and Methanol Extracts, and their Respective Fractions

Briefly, the SOR was chopped into small pieces and ground to a fine powder using a blender. The powder (50 g) was extracted for 24 h with 500 mL of cold water (4 °C), 2 h with 500 mL of hot water (120 °C, autoclaved), or 24 h with 500 mL of methanol in a shaking incubator. All aliquots of each extract were filtered using Whatman No. 1 filter paper (GE Healthcare, Buckinghamshire, UK), reduced to 10 mL by a vacuum rotary evaporator (EYELA N-N, Tokyo, Japan) at 60 °C, and lyophilized for 4 days to obtain dried powder. The CWE, HWE, and ME yields were 1.8, 3.9, and 2.8 g, respectively, for use as samples after dissolving in dimethyl sulfoxide (DMSO). Furthermore, each dried powder was dissolved in 50 mL of 50% methanol and re-extracted successively with equal volumes of n-hexane (fraction 1), EA (fraction 2), *n*-butanol (fraction 3), and water (fraction 4) in order. Among the four fractions, the EA fractions of the respective extracts exhibited the most active antibacterial activity (data not shown) against *P. acnes* based on the paper disc diffusion method described below. Therefore, we excluded the other fractions from the subsequent experiments. The respective EA fractions were evaporated at low temperature under reduced pressure, freeze-dried and powdered, and dissolved in DMSO for use as samples. 

### 4.3. Analysis of the CWE, HWE, ME, and their Respective EA Fractions

#### 4.3.1. Determination of Total Phenolic Content

The total phenolic content in the CWE, HWE, and ME from SOR, and their respective EA fractions, was determined using the Folin–Ciocalteu method [59], with a minor modification. For preparation of the calibration curve, 50 μL aliquots of 0.024, 0.075, 0.105, and 0.3 mg/mL of methanolic gallic acid solution were mixed with 500 μL of 10% Folin–Ciocalteau’s reagent in water and 400 μL of 1 M sodium bicarbonate. After 30 min of treatment in the dark, the absorption was read at 765 nm at 20 °C, and the calibration curve was drawn. Fifty microliters of each sample (1 mg/mL in methanol) was mixed with the same reagents described above. A reagent blank was also prepared using methanol. After 30 min, the absorption was measured for determination of plant phenolics. The total phenolic content was expressed as gallic acid equivalent (GAE) milligrams per gram of dry powder. All determinations were performed in triplicate.

#### 4.3.2. Determination of Total Flavonoid Content

The total flavonoid content in the CWE, HWE, and ME from SOR, and their respective EA fractions, was determined by Moreno’s method [60], with slight modifications. Each sample (20 μL of 1 mg/mL in methanol) of the CWE, HWE, ME, and their respective EA fractions was mixed with 20 μL of 10% (*w*/*v*) aluminum nitrate, 4 μL of 1 M potassium acetate, 60 μL of methanol, and 112 μL of distilled water. The mixture was kept at room temperature for 30 min, and then its absorption at 415 nm was read using a UV-VIS spectrophotometer Libra S22 (Biochrom Ltd., Cambridge, UK). A standard curve was prepared by measuring the absorption of quercetin solutions in methanol (0–100 μg/mL) under the same conditions. The total flavonoid content was expressed as mg of quercetin equivalent (QE)/g of dry powder. All determinations were performed at least in triplicate. 

#### 4.3.3. Determination of Total Terpenoid Content

The total terpenoid content in the CWE, HWE, and ME from SOR, and their respective EA fractions, was determined according to the method of Ghorai et al. [61] using linalool as a standard reagent. To 200 μL of each sample (1 mg/mL in methanol) and 1.5 mL of chloroform was added. The mixture was vortexed thoroughly and kept for 3 min at room temperature, then 100 μL of concentrated sulfuric acid was added. The microcentrifuge tube containing the reaction mixture was incubated at room temperature for 1.5 h in the dark. When a reddish brown precipitate formed, the supernatant reaction mixture was gently decanted without disturbing the precipitate. After adding 1.5 mL of methanol (95%, *v*/*v*) into the microcentrifuge tube, the precipitate was completely dissolved by vortexing, and the resulting mixture was transferred to a colorimetric cuvette to read the absorbance at 538 nm against methanol as a blank. The total terpenoid content of each sample was calculated as Linalool equivalents (mg/g) using the regression equation of the Linalool standard curve (y = 0.012 x + 0.011, r^2^ = 0.982). All determinations were performed at least in triplicate.

#### 4.3.4. Determination of Condensed Tannin Content

The condensed tannin content in the CWE, HWE, and ME from SOR, and their derived EA fractions, was determined according to the method of Sun et al. [62], with minor modifications using catechin as a reference compound. To 400 μL of each sample (1 mg/mL in methanol), 3 mL of 4% vanillin solution in methanol and 1.5 mL of concentrated HCl were added. The mixture was allowed to stand for 15 min at room temperature, and the absorption was measured at 500 nm against methanol as a blank. The amount of total condensed tannins was expressed as mg of catechin equivalent (CE)/g of dry powder. All determinations were performed at least in triplicate.

### 4.4. Microorganisms and Culture

The Gram-negative (Escherichia coli and Vibrio parahaemolyticus) and Gram-positive (Listeria monocytogenes, Staphylococcus aureus, and *P. acnes*) bacteria were used as the test strains. *E. coli*, *V. parahaemolyticus*, *L. monocytogenes*, and *S. aureus* were incubated in LB medium at 37 °C for 24 h in a CO_2_ incubator, whereas *P. acnes* was incubated in brain-heart infusion (BHI) medium for 48 h at 37 °C under anaerobic conditions in an anaerobic jar (Mitsubishi Gas Chemical Co., Tokyo, Japan) with a gas pack.

### 4.5. Determination of Antibacterial Activity

#### 4.5.1. Paper Disc Diffusion Method

The antibacterial activities of the samples were initially evaluated by the disc diffusion assay [63]. A sterile cotton swab was dipped into overnight bacterial suspensions of the test strains and used to inoculate over the selective agar (1.5%) medium by evenly streaking. Stock solutions of the CWE, HWE, ME, and their respective EA fractions were prepared in dimethyl sulfoxide (DMSO) at a concentration of 10 mg/mL and diluted to the concentrations required for the treatments. Paper filter discs (8 mm) impregnated with 20 μL (1, 3, or 5 mg/disc) of each sample were separately placed on the medium surface. The plates were left for 30 min at room temperature to allow diffusion of the extracts, and then incubated at 37 °C for 24 h for *E. coli*, *V. parahaemolyticus*, *L. monocytogenes*, and *S. aureus*, and 48 h anaerobically for *P. acnes*. Finally, the diameters of the inhibition zones around the disc were measured. Kanamycin was included in the test as a reference control to evaluate the susceptibility of the tested strains. The experiments were run in triplicate.

#### 4.5.2. MIC and MBC

The *P. acnes* bacteria were prepared at 48 h BHI broth cultures, and the concentration was adjusted to 0.5 OD_600nm_. A diluted bacterial suspension (100 μL) was inoculated into each well of a 96-well microplate. The MIC was determined in μg/mL for the CWE, HWE, ME, and their respective EA fractions using a two-fold serial dilution assay. Each sample was diluted in DMSO to a concentration of 5000 μg/mL, and serial dilutions were made to obtain a concentration range from 5000 to 19.5 μg/mL. A diluted sample (100 μL) was added to each well of the microplate. A medium blank with the selective broth and the sample solution was also prepared for the controls. The MBC was determined by subculturing 100 μL of the samples on sterile BHI agar plates from 3 wells that showed no growth during the MIC determination. The plates were incubated following the procedure described for the MIC determination. The MBC was interpreted as the lowest concentration of the sample that showed no growth on the agar plates. 

### 4.6. In Vitro Antioxidant Activities of the CWE, HWE, ME, and their Respective EA Fractions

#### 4.6.1. DPPH Assay

Each methanolic sample (1 mg/mL in methanol) was further diluted using the two-fold method and subjected to the DPPH radical scavenging assay according to the method of Choi et al. [64], with slight modifications. Each diluted sample (20 μL) was mixed with 0.2 mM DPPH (180 μL) dissolved in methanol, and the mixture was allowed to react in the dark for 15 min at room temperature. Methanolic DPPH solution without a sample was used as a control. The absorbance was measured against a blank at 517 nm and converted into the percent SC using the following equation: % SC = (Absorbance of control − Absorbance of sample) × 100/Absorbance of control. The SC_50_ value was calculated by linear regression of the plots and was defined as the concentration of sample required to reduce 50% of the DPPH free radicals. All tests were performed at least in triplicate, and the graphs were plotted using the average of three determinations.

#### 4.6.2. Total Antioxidant Assay

The total antioxidant activity of the methanolic samples was determined as described by Shabbir et al. [65], with slight modifications. Each sample (100 μL of 1 mg/mL sample in methanol) was mixed with 600 μL of reagent (0.6 M sulfuric acid, 28 mM sodium phosphate, and 4 mM ammonium molybdate). The mixture was incubated at 95 °C for 15 min in a water bath and then cooled to room temperature. The absorbance was measured at 765 nm using a UV-VIS spectrophotometer against a reagent blank. Quercetin (1 mg/mL) in methanol was used as a reference, and its total antioxidant activity was defined as 100%. The total antioxidant activity of each sample was expressed relative to that of quercetin. All determinations were performed in triplicate.

#### 4.6.3. Hydroxide Peroxide Scavenging Assay

The hydrogen peroxide (H_2_O_2_) SC of the methanolic samples was determined according to the method of Hazra et al. [66]. Each sample (100 μL of a 1 mg/mL sample in methanol) was mixed with 100 μL of hydrogen peroxide (50 mM) and incubated for 30 min at room temperature. The sample mixture (90 μL) was made up to 100 μL with HPLC-grade methanol, and 0.9 mL of the FOX reagent (9 volumes of 4.4 mM butylated hydroxytoluene in HPLC-grade methanol: 1 volume of 1 mM xylenol orange and 2.56 mM ammonium ferrous sulfate in 250 mM H_2_SO_4_) was added. The total reaction mixture was vortexed and incubated at room temperature for 30 min. The absorbance of the ferric-xylenol orange complex was measured at 560 nm. Quercetin (1 mg/mL) in methanol was used as a reference, and its SC was defined as 100%. The SC of each sample was expressed relative to that of quercetin. All determinations were performed in triplicate.

#### 4.6.4. Linoleic Acid Peroxidation Inhibition Assay

The antioxidant activity of the methanolic samples was also determined in terms of percent inhibition of linoleic acid peroxidation with the following method. Briefly, each sample (200 μL of a 1 mg/mL sample in methanol) was added to a solution containing 100 μL of linoleic acid (10 mM), 100 μL of FeSO_4_ (10 μM), and 100 μL of ascorbic acid (2 mM), and the reaction mixture was incubated at 37 °C for 1 h. The reaction was terminated by adding 100 μL of TCA (28%) and 300 μL of thiobarbituric acid (TBA, 1%), followed by incubation at 80 °C for 1 h. Lipid peroxide reacts with TBA to form thiobarbituric acid reactive substances (TBARS); its absorbance at 532 nm was measured to quantify the TBARS. Quercetin was used as a standard for comparison. 

### 4.7. HPLC Analysis

The lyophilized fractions of the CWE-EA, HWE-EA, and ME-EA were reconstituted in 1 mL of 20% aqueous methanol (*v*/*v*) and passed through a 0.2 μm nylon filter. Eleven standards were included for flavonoids (quercetin and kaempferol), triterpenoids (arjunic acid and euscaphic acid), polyphenols (caffeic acid, chlorogenic acid, ferulic acid, gallic acid, tannic acid, and coumarin), and kojic acid. Chromatographic separations were performed on an Agilent Zorbax Eclipse XDB-C18 (Agilent, Santa Clara, CA, USA) column (4.6 mm × 150 mm, 5 μm). Samples (10 μL) were injected into the HPLC instrument (Shimadzu Prominence LC 20A series HPLC system, Shimadzu Corp, Kyoto, Japan) with a PDA detector. The mobile phase for CWE-EA, HWE-EA, and ME-EA consisted of 0.1% phosphoric acid in water (solvent A) and 0.1% phosphoric acid in acetonitrile (solvent B). Elution from the column was achieved with the following gradient: 0–5 min, 97% A and 3% B; 15–20 min, 90% A and 10% B; 30–40 min, 50% A and 50% B; 40.1–50 min, 97% A and 3% B. The preparative system was run for 40 min of the total running time at a constant flow rate of 0.8 mL/min at ambient temperature, and the spectrum was monitored at 272 nm. The identification of each compound was based on a combination of the retention time and UV spectral matching. 

### 4.8. UHPLC-QTOF-MS

The mass spectrometry system was a 1290 Infinity II Ultra High Performance Liquid Chromatography (Agilent, Santa Clara, CA, USA) Triple TOF 5600 plus time of flight mass spectrometer, equipped with electrospray ionization (ESI) source TripleTOF^®^ 5600+ (AB SCIEX, Framingham, MA, USA). The mobile phase for CWE-EA and HWE-EA consisted of 2% acetic acid in water (solvent A) and 2% acetic acid in 50% acetonitrile (solvent B), whereas the mobile phase for ME-EA consisted of 2% acetic acid in water (solvent A) and 2% acetic acid in 100% methanol (solvent B). The MS analysis was performed by ESI positive ion scanning mode. The conditions of the ESI source were as follows: Drying gas flow rate, 10 L/min; drying gas temperature, 400 °C; sheath gas flow rate, 10 L/min; sheath gas temperature, 400 °C; nebulizer, 45 psi; capillary voltage, 4500 V; fragmentor voltage, 180 V; mass range of 100–1000 *m*/*z*; scan rate, 3 Hz. The resolution was 35,000 FWHM. Data acquisition and processing were done using PeakView^®^ 2.2 and MasterView™ 1.1 (AB SCIEX, Framingham, MA, USA). Confidence of the compound identification was based on accurate mass, error mass (+/− 3.0 ppm), and isotope difference (<20%), and expressed by an overall identification score computed as a weighted average of the compound isotopic signals, such as exact mass, relative abundances, and *m*/*z* distances. To determine the anti-*P. acnes* activity of the identified compounds, nine standards were tested using the paper disc diffusion method described above. 

### 4.9. Statistical Analysis

The statistical analysis was performed using one-way analysis of variance (ANOVA) followed by Duncan’s multiple range test in the SPSS software (version 19.0, SPSS Inc., Chicago, IL, USA) (*p* < 0.05). Pearson’s correlation coefficients (http://www.socscistatistics.com) were used to determine whether the antibacterial activity or antioxidant activity was associated with the total phenolic, flavonoid, terpenoid and condensed tannin contents (*p* < 0.01 or *p* < 0.05).

## 5. Conclusions

The present investigation demonstrated that the SOR CWE, HWE, ME, and their respective EA fractions had strong anti-*P. acnes* and antioxidant activities. The majority of phenolic acids, flavonoids, and tannins were identified in all EA fractions, and were considered to be responsible for the antioxidant activity of SOR. On the other hand, triterpenoids (euscaphic acid) combined with quercetin (HWE-EA) or coumarin (ME-EA) could be responsible for the strong anti-*P. acnes* activity of SOR.

## Figures and Tables

**Figure 1 molecules-23-03001-f001:**
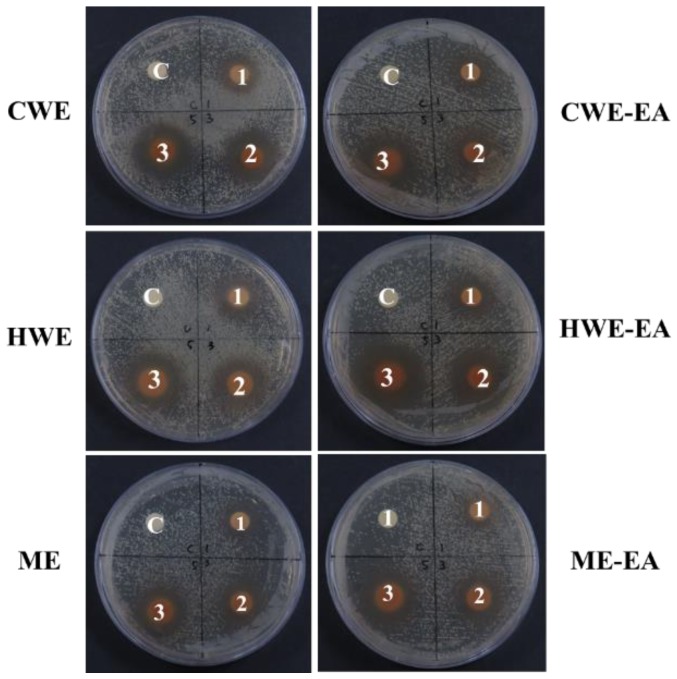
Determination of inhibition zones against *P. acnes* using the paper disc diffusion method on agar plates. The treated concentrations were 1 mg/mL (**1**), 3 mg/mL (**2**), and 5 mg/mL (**3**) of the *S. officinalis* L. root (SOR) CWE, HWE, ME, and their derived ethyl acetate fractions (CWE-EA, HWE-EA, and ME-EA). The negative control (**C**) in each sample is DMSO.

**Figure 2 molecules-23-03001-f002:**
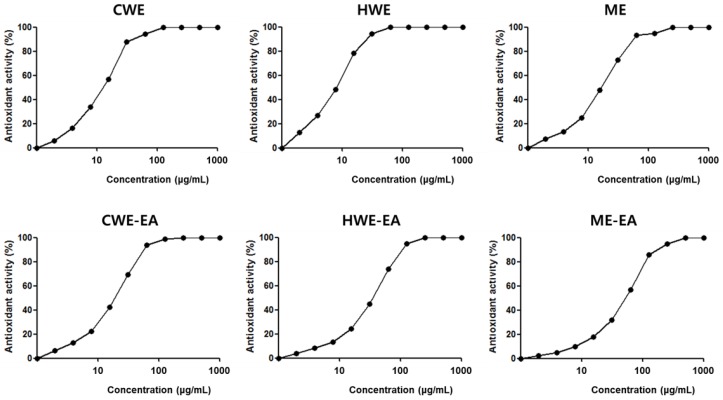
Scavenging capacity (SC) of the 2,2-diphenyl-1-picrylhydrazyl (DPPH) radical by the SOR CWE, HWE, ME, and their ethyl acetate fractions (CWE-EA, HWE-EA, and ME-EA). Each methanolic sample (1 mg/mL in methanol) was further diluted using the two-fold method. The absorbance was measured against a blank at 517 nm and converted into the percentage SC using the following equation: %SC = (Absorbance of control − Absorbance of sample) × 100/Absorbance of control. All tests were performed at least in triplicate, and the graphs were plotted using the average of three determinations.

**Figure 3 molecules-23-03001-f003:**
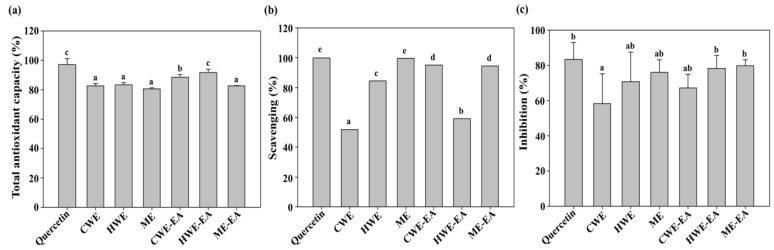
In vitro antioxidant activities of the SOR CWE, HWE, ME, and their respective ethyl acetate fractions (CWE-EA, HWE-EA, and ME-EA) at 1 mg/mL concentration. (**a**) The total antioxidant activity was determined spectrophotometrically at 765 nm. (**b**) The H_2_O_2_ scavenging capacity (%) was determined spectrophotometrically at 560 nm. (**c**) The lipid peroxidation inhibition (%) was determined by the thiobarbituric acid method at 535 nm. All data is expressed as the mean ± SD (*n* = 3). Values with the same letter on each bar are not significantly different using Duncan’s multiple range test at the 5% level (*p* < 0.05).

**Figure 4 molecules-23-03001-f004:**
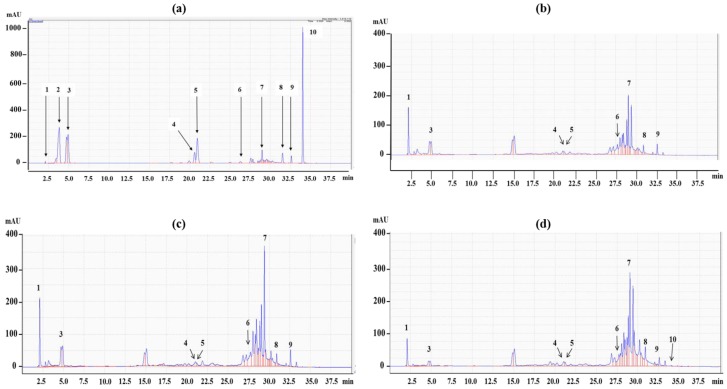
High performance liquid chromatography-ultraviolet (HPLC-UV) chromatograms (272 nm) of the mixed references solution (**a**), SOR CWE-EA (**b**), HWE-EA (**c**), and ME-EA (**d**). The numbers represent euscaphic acid and/or arjunic acid (**1**), comic acid (**2**), gallic acid (**3**), kaempferol (**4**), caffeic acid (**5**), ferulic acid (**6**), tannic acid (**7**), coumarin (**8**), quercetin (**9**), and chlorogenic acid (**10**).

**Figure 5 molecules-23-03001-f005:**
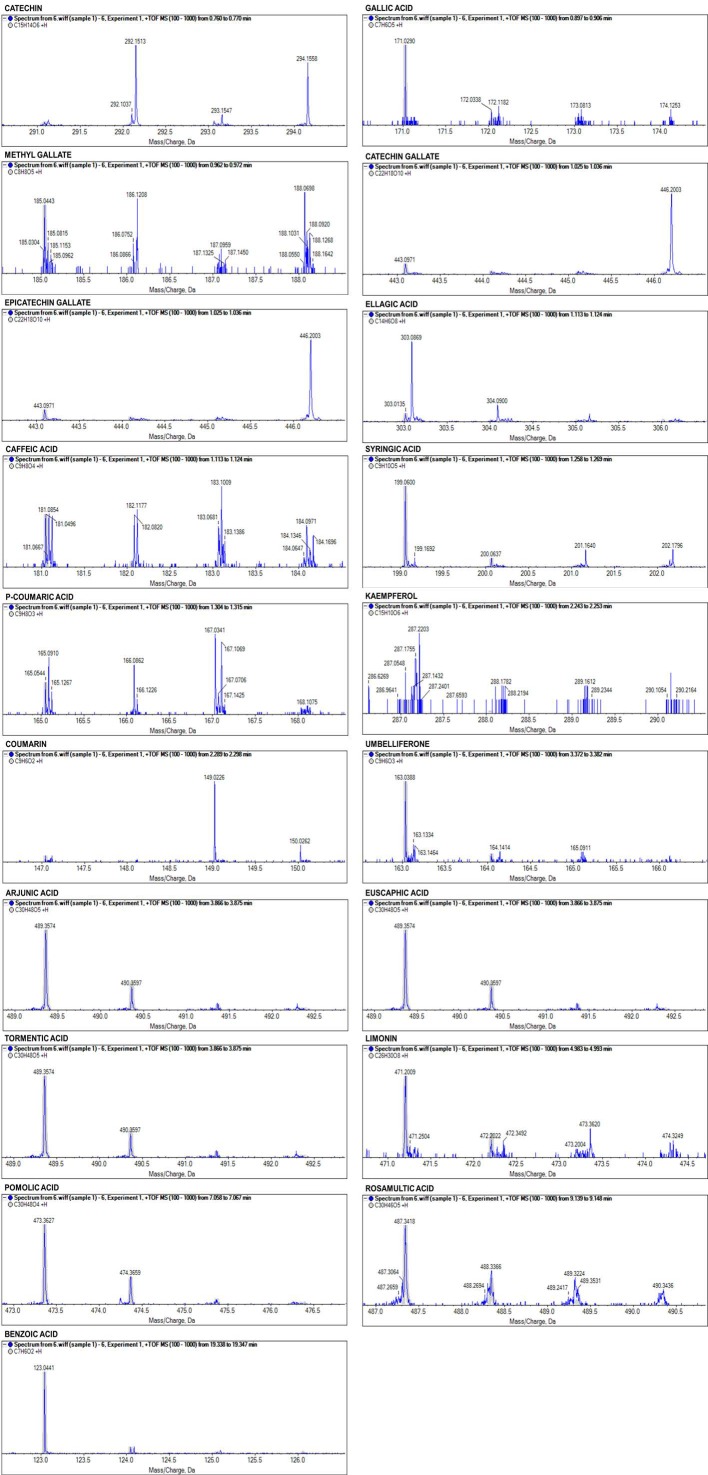
UHPLC-QTOF-MS spectra of major compounds in SOR ME-EA.

**Figure 6 molecules-23-03001-f006:**
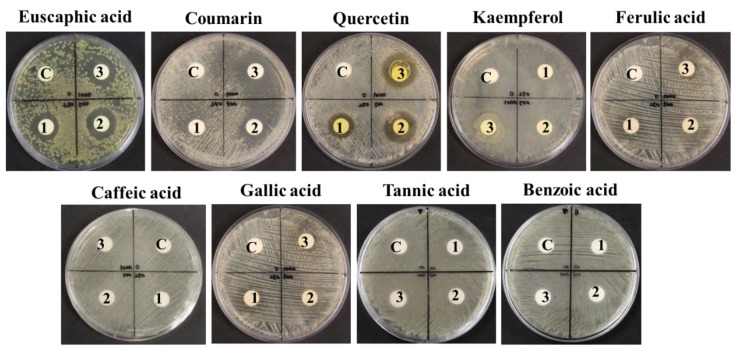
Effects of HPLC-identified compounds on the growth inhibition of *P. acnes* using the paper disc diffusion method on agar plates. The concentrations of the treated standard chemicals were DMSO-treated control (**C**), 250 μg/mL (**1**), 500 μg/mL (**2**), and 1000 μg/mL (**3**).

**Table 1 molecules-23-03001-t001:** Total phenolic, flavonoid, terpenoid, and condensed tannin contents in various extracts and their respective fractions of *S. officinalis* L. roots.

Extract/Fraction	Total Phenolic Content (mg GAE/g ± SD)	Total Flavonoid Content (mg QE/g ± SD)	Total Terpenoid Content (mg/g ± SD)	Condensed Tannin Content (mg CE/g ± SD)
CWE	59.0 ± 0.57 ^d^	20.6 ± 0.39 ^ab^	29.10 ± 0.000 ^c^	1.69 ± 0.058 ^b^
HWE	60.0 ± 1.15 ^d^	30.2 ± 2.46 ^c^	27.40 ± 0.001 ^b^	1.80 ± 0.004 ^c^
ME	46.1 ± 3.23 ^c^	17.9 ± 0.81 ^a^	22.38 ± 0.001 ^a^	1.14 ± 0.001 ^a^
CWE-EA	58.1 ± 1.89 ^d^	20.9 ± 0.34 ^ab^	29.69 ± 0.001 ^c^	2.15 ± 0.002 ^e^
HWE-EA	38.3 ± 2.41 ^b^	20.1 ± 0.92 ^ab^	26.29 ± 0.001 ^b^	2.06 ± 0.001 ^ab^
ME-EA	25.6 ± 1.99 ^a^	18.2 ± 2.31 ^ab^	29.68 ± 0.001 ^c^	1.73 ± 0.007 ^ab^

CWE: cold water extract; HWE: hot water extract; ME: methanol extract; CWE-EA: ethyl acetate fraction of the CWE; HWE-EA: ethyl acetate fraction of the HWE; ME-EA: ethyl acetate fraction of the ME. GAE: gallic acid equivalent; QE: quercetin equivalent; CE: catechin equivalent. g: dry powder weight. The data shown represent the mean values of triplicate assays and standard deviations (SDs). Values in the same column followed by a different letter are significantly different using Duncan’s multiple range test at the 5% level (*p* < 0.05).

**Table 2 molecules-23-03001-t002:** Minimum inhibitory concentration (MIC) and minimum bactericidal concentration (MBC) against *P. acnes* of various extracts from *S. officinalis* L. and their ethyl acetate fractions.

Extract & Fraction	MIC (μg/mL)	MBC (μg/mL)
CWE	1250	2500
HWE	625	1250
ME	312	312
CWE-EA	312	312
HWE-EA	156	312
ME-EA	78	156

**Table 3 molecules-23-03001-t003:** MS data and the identification of compounds from CWE-EA, HWE-EA and ME-EA by UHPLC-QTOF-MS.

Sample	Compound Name	Score (%)	Formula	Intensity	Expected/Found (*m*/*z*)	Error (ppm)	RT (min)
CWE-EA	Gallic acid	93	C_7_H_6_O_5_	10,437	171.0288/171.0287	−0.7	0.91
	Myricetin	93	C_15_H_10_O_8_	2638	319.0448/319.0447	−0.5	1.06
	Caffeic acid	94	C_9_H_8_O_4_	5899	181.0495/181.0494	−0.7	1.15
	Trans-ferulic acid	92	C_10_H_10_O_4_	3584	195.0652/195.0652	−0.1	1.33
	Umbelliferone	93	C_9_H_6_O_3_	8637	163.0390/163.0388	−0.9	3.38
	Arjunic acid	92	C_30_H_48_O_5_	18,571	489.3575/489.3571	−0.7	3.88
	Euscaphic acid	92	C_30_H_48_O_5_	18,571	489.3575/489.3571	−0.7	3.88
	Tormentic acid	92	C_30_H_48_O_5_	18,571	489.3575/489.3571	−0.7	3.88
	Pomolic acid	94	C_30_H_48_O_4_	56,259	473.3625/473.3623	−0.6	3.88
	Rosamultic acid	54	C_30_H_46_O_5_	14,695	487.3418/487.3415	−0.7	8.24
	Oleic acid	94	C_18_H_34_O_2_	55,293	283.2632/283.2630	−0.7	12.27
	Benzoic acid	96	C_7_H_6_O_2_	33,141	123.0441/123.0440	−0.4	19.92
HWE-EA	Catechin	51	C_15_H_14_O_6_	3749	291.0863/291.0864	0.4	0.73
	Gallic acid	61	C_7_H_6_O_5_	8040	171.0288/171.0288	−0.1	0.92
	Hyperoside	66	C_21_H_20_O_12_	2389	465.1028/465.1032	1.0	1.00
	Catechin gallate	79	C_22_H_18_O_10_	2560	443.0973/443.0975	0.6	1.06
	epicatechin gallate	79	C_22_H_18_O_10_	2560	443.0973/443.0975	0.6	1.06
	Ellagic acid	77	C_14_H_6_O_8_	9047	303.0135/303.0136	−0.2	1.07
	Caffeic acid	92	C_9_H_8_O_4_	4961	181.0495/181.0496	0.6	1.15
	Syringic acid	88	C_9_H_10_O_5_	16,304	199.0601/199.0599	−0.9	1.25
	Trans-ferulic acid	74	C_10_H_10_O_4_	4102	195.0652/195.0650	−0.7	1.30
	Umbelliferone	92	C_9_H_6_O_3_	9133	163.0390/163.0388	−1.0	1.93
	Quercetin	87	C_15_H_10_O_7_	2978	303.0499/303.0498	−0.4	2.35
	Arjunic acid	64	C_30_H_48_O_5_	14,370	489.3575/489.3577	0.5	3.89
	Euscaphic acid	64	C_30_H_48_O_5_	14,370	489.3575/489.3577	0.5	3.89
	Tormentic acid	64	C_30_H_48_O_5_	14,370	489.3575/489.3577	0.5	3.89
	Kaempferol	85	C_15_H_10_O_6_	1929	287.0550/287.0551	0.4	3.90
	Limonin	77	C_26_H_30_O_8_	14,017	471.2013/471.2001	−2.7	4.99
	Pomolic acid	94	C_30_H_48_O_4_	29,780	473.3625/473.3622	−0.7	7.06
	Rosamultic acid	57	C_30_H_46_O_5_	9707	487.3418/487.3420	0.3	8.25
	Benzoic acid	93	C_7_H_6_O_2_	40,741	123.0441/123.0439	−1.0	20.30
ME-EA	Catechin	50	C_15_H_14_O_6_	5571	291.0863/291.0865	0.8	0.73
	Gallic acid	55	C_7_H_6_O_5_	9194	171.0288/171.0290	1.0	0.88
	Methyl gallate	86	C_8_H_8_O_5_	3047	185.0444/185.0443	−1.0	0.95
	Catechin gallate	74	C_22_H_18_O_10_	4935	443.0973/443.0971	−0.5	1.04
	Epicatechin gallate	74	C_22_H_18_O_10_	4935	443.0973/443.0971	−0.5	1.04
	Ellagic acid	96	C_14_H_6_O_8_	4378	303.0135/303.0135	−0.1	1.07
	Caffeic acid	96	C_9_H_8_O_4_	2066	181.0495/181.0496	0.4	1.07
	Syringic acid	95	C_9_H_10_O_5_	31,460	199.0601/199.0600	−0.7	1.26
	*p*-coumaric acid	60	C_9_H_8_O_3_	1944	165.0546/165.0544	−1.6	1.36
	Kaempferol	81	C_15_H_10_O_6_	2560	287.0550/287.0548	−0.7	2.25
	Coumarin	84	C_9_H_6_O_2_	4101	147.0441/147.0442	1.1	2.27
	Umbelliferone	93	C_9_H_6_O_3_	9469	163.0390/163.0388	−0.9	3.36
	Arjunic acid	74	C_30_H_48_O_5_	26,745	489.3575/489.3574	−0.1	3.85
	Euscaphic acid	74	C_30_H_48_O_5_	26,745	489.3575/489.3574	−0.1	3.85
	Tormentic acid	74	C_30_H_48_O_5_	26,745	489.3575/489.3574	−0.1	3.85
	Limonin	56	C_26_H_30_O_8_	11,658	471.2013/471.2099	−1.0	4.99
	Pomolic acid	97	C_30_H_48_O_4_	92,518	473.3625/473.3627	0.4	7.06
	Rosamultic acid	55	C_30_H_46_O_5_	55,933	487.3418/487.3418	0.1	9.29
	Benzoic acid	97	C_7_H_6_O_2_	37,983	123.0441/123.0441	0.4	19.32

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
