# Peer review of "In Vitro Antioxidant and Anti-Propionibacterium acnes Activities of Cold Water, Hot Water, and Methanol Extracts, and Their Respective Ethyl Acetate Fractions, from Sanguisorba officinalis L. Roots"

_molecules, 2018, doi:10.3390/molecules23113001_

Reviewer 1 Report

The authors have performed a study on antioxidant and anti-propionibacterium acnes activity of varios extracts of Sanguisorba officinalis L. roots. The work is interesting but the novelty and reason for doing this is not clearly defined. The main standout results are also not well expressed either and this needs improvement. After corrections this may be suitable for Molecules.

Table 1: The significant differences are very difficult to follow in this table for example total phenolic content column there is only 1 a so how is that significant to anything else in that column? If comparing between columns this seems pointless as you would expect them to be different anyway.

Figure 1: Were replicates done of this and just showing a representative figure?

Page 9: lines 190-194: Just to be clear are the authors saying that the coumarin was the most active and detected in only the ME-EA extracts? This seems the be the most important point of the manuscript as it explains why the activity is greatest with the ME-EA extracts.

Page 10 lines 268-269: The area of the peak is proportional is a much more accurate statement. Any analysis using height should be replaced for area to check for differences.

Page 14 lines 430-433: The name of the mass spectrometer is TripleTOF® 5600+ can the method be corrected to this (+ not plus). It is also worth stating that it’s a QTOF as the triple TOF name can be misleading to those who don’t know, Throughout the manuscript UHPLC-Triple-TOF-MS needs to be corrected to UHPLC-QTOF-MS.

Author Response

We have followed the reviewer's commentsto revise our manuscript or make a suitable rebuttal. Thank you for the valuable comments. We agree that these comments have vastly improved our manuscript.

Reviewer 2 Report

page 1 line 24: correct the chemical name of DPPH

page 1 line 30: change to “were performed”

page 6 lines 172 and 174: error mass or mass error?

page 10 line 269: correct “each peaks”

page 10 line 302: change to “were mixed”

Author Response

We have followed the reviewer's comments to revise our manuscript or make a suitable rebuttal. Thank you for the valuable comments. We agree that these comments have vastly improved our manuscript. 
